

# Photovoltaic panel defect detection algorithm based on infrared imaging and improved YOLOv8

Jingdong Wang and Zhu Cheng

School of Computer Science, Northeast Electric Power University, Jilin, Jilin, China

## ABSTRACT

To address the challenges of high missed detection rates, complex backgrounds, unclear defect features, and uneven difficulty levels in target detection during the industrial process of photovoltaic panel defect detection, this article proposes an infrared detection method based on computer vision, with enhancements built upon the YOLOv8 model. First, a multi-channel squeeze-and-excitation network is introduced to improve feature extraction capabilities and is integrated into the neck network. Second, GhostConv and BoTNet are incorporated into the backbone network to reduce model parameters while enhancing defect detection performance. Finally, the Focaler-Complete Intersection over Union (Focaler-CIoU) loss function is employed to tackle the issue of imbalanced difficulty in target detection tasks. The method is evaluated on the PV-Multi-Defect-main dataset and further validated through a generalization test on the PVEL-AD dataset. Results demonstrate that, compared with the baseline YOLOv8 model, the proposed approach achieves significant improvements in precision (3.6%), recall (10.4%), mAP50 (4.8%), and mAP50-95 (4.5%) while maintaining nearly the same parameter count. On the PVEL-AD dataset, the method effectively addresses the challenge of feature extraction failure for dislocation-type defects, achieving substantial gains in precision (7.8%), recall (17.1%), mAP50 (19.5%), and mAP50-95 (13.2%). Furthermore, comparisons with several state-of-the-art detection algorithms reveal that the proposed method consistently delivers improved detection performance, validating its effectiveness as a robust solution for photovoltaic panel defect detection.

## INTRODUCTION

Surface defect detection of photovoltaic (PV) panels is of significant practical importance for improving power generation efficiency and reducing safety risks. Traditional detection methods such as penetration testing (*Osa et al., 2015*), X-ray inspection (*Pan et al., 2021*), and acoustic testing (*Choudhury & Nandi, 2023*) have limitations in terms of practicality. For instance, these methods often suffer from high equipment costs, low detection accuracy, and limited defect type coverage. Penetrant testing visualizes defects on photovoltaic panels by applying penetrant liquid to the surface and using a developer to make the defects visible. However, this method can only detect surface or near-surface cracks, involves cumbersome procedures, and leaves residual penetrant liquid that can

Corresponding author
Zhu Cheng,
czzc898726506@163.com

damage both the photovoltaic panel and the environment. X-ray testing identifies internal defects by utilizing the differences in X-ray absorption between various materials. X-rays can reveal internal structural defects such as voids, inclusions, and cracks. However, the high cost of X-ray inspection equipment limits its application in the operation and maintenance of photovoltaic power stations. Additionally, X-ray images may contain noise due to uneven radiation penetration, requiring robust denoising and feature extraction capabilities during image processing. In ultrasonic testing (*Zhu et al., 2017*), the analysis of acoustic signals is challenging and does not directly provide the exact location and size of defects, limiting its flexibility and applicability.

In an era of rapid advancements in artificial intelligence and the booming growth of the renewable energy industry, detecting defects in PV panels accurately and effectively using infrared imaging based on the principle of electroluminescence holds immense practical value. This approach has the potential to replace manual inspection and other existing industrial detection methods, making it a promising candidate to become the primary method for PV panel defect detection (*Qu, Jiang & Zhang, 2020*). In recent years, the trend of leveraging computer vision technology to achieve this goal has become increasingly evident, marking a critical direction for artificial intelligence applications in the power industry. Researchers have proposed a variety of targeted algorithms for PV panel defect detection (*Wang et al., 2022*; *Di Tommaso et al., 2022*; *Li, Wang & Zhang, 2023*; *Lee, Yan & Yang, 2023*; *Cao et al., 2023*; *Zhao et al., 2023*).

However, the diversity of defect types poses significant challenges. For example, although infrared-based detection methods (*Bu et al., 2023*) are cost-effective and fast, they are not capable of effectively detecting a wide range of defect types. Existing detection algorithms often lack generalizability in accurately identifying all types of defects, as many researchers focus on specific types of defects, thereby limiting their practical application. This limitation arises because infrared signals originate from the thermal radiation of the PV panel surface. For certain defects—especially those that are small or have low local resistance—the temperature difference between the defect and its surrounding area is minimal, making them difficult to detect using infrared cameras. Consequently, infrared detection methods are primarily effective for defect types with significant temperature differences, such as hotspots.

In the field of PV panel defect detection, although the original YOLOv8 network offers numerous advantages, it still has certain limitations. First, PV panels are typically installed in outdoor environments with complex and dynamic backgrounds. If there are features or textures in the background that resemble defects such as cracks, broken grids, or scratches, YOLOv8 may be susceptible to interference, leading to reduced detection accuracy. Second, as a general-purpose object detection network, YOLOv8 lacks optimization for electroluminescence (EL) defect images of PV panels, which limits its detection performance. For example, scratches, being small-object defects, and defects like black_core, which have indistinct boundaries, are challenging to detect accurately. The low precision of the model fails to meet the requirements of industrial inspection. Third, increasing model size (*e.g.*, using larger variants such as YOLOv8-m or YOLOv8-l) to

improve detection accuracy results in higher model complexity, making real-time on-site industrial inspections impractical. Therefore, the model should aim to maximize detection accuracy while maintaining a lightweight design to meet the demands of industrial-level defect detection.

To address these limitations (*Hussain & Khanam, 2024*), this study proposes a PV panel defect detection method based on YOLOv8 and computer-based infrared vision. We modify the YOLOv8 algorithm to optimize its performance and enhance its applicability in industrial inspection scenarios. Our improved model is designed to increase the accuracy and robustness of defect detection in PV panels. Specifically, the major contributions of this research are as follows:

1) A novel multi-channel squeeze-and-excitation (MCSE) attention mechanism: We propose a new MCSE attention mechanism to overcome a fundamental limitation of traditional convolution operations, which typically perform convolutions within individual channels and then sum the results. This process often leads to suboptimal mixing of spatial and channel features. The MCSE mechanism enables the model to directly learn channel-specific features, significantly enhancing its ability to extract and identify defect-related characteristics. In object detection, this attention mechanism allows the model to focus more on critical regions of the image while ignoring background noise.

2) Architectural improvements to YOLOv8: To improve the efficiency of defect detection in PV panels, we made several enhancements to the YOLOv8 architecture. Specifically, we introduced GhostConv to replace a portion of the original convolution layers, reducing computational complexity while maintaining feature representation. Additionally, we integrated BoTNet to strengthen the backbone network's feature extraction capabilities. Furthermore, we replaced the standard CIoU loss function with a novel Focaler-CIoU loss, designed to optimize model performance while preserving computational efficiency.

3) A Defect detection model for PV panel electroluminescence images: We developed a defect detection model tailored to EL images of PV panels, addressing the poor detection performance of the original YOLOv8 network in industrial applications. Our model achieves improved detection accuracy while maintaining nearly the same parameter size, thereby enhancing its industrial applicability.

This study demonstrates that the proposed modifications effectively improve YOLOv8's performance and usability in detecting defects in PV panels, paving the way for more accurate and efficient industrial inspection solutions.

# RELATED WORK

## Object detection

In the field of computer vision-based photovoltaic panel defect detection, algorithms can be broadly divided into two main categories: single-stage and two-stage models.

Two-stage models operate through a sequential process. First, they generate multiple region proposals from the input image. Next, features are extracted from each proposed region and transformed into uniform-sized feature maps using region of interest (RoI) pooling. In the final stage, these models perform classification and bounding box regression, enabling precise predictions of object labels and their corresponding bounding boxes. Notable examples of two-stage algorithms include fast region-based convolutional neural network (Fast R-CNN) (*Girshick, 2015*) and mask region-based convolutional neural network (Mask R-CNN) (*He et al., 2017*).

In contrast, single-stage models eliminate the region proposal step entirely. Instead, they integrate feature extraction, classification, and bounding box localization within a fully convolutional network architecture. This streamlined design significantly accelerates processing times compared to two-stage models, making single-stage approaches particularly well-suited for real-time applications. Prominent examples of single-stage algorithms include single-shot detector (SSD) (*Liu et al., 2016*) and the You Only Look Once (YOLO) series (*Redmon et al., 2016*; *Redmon & Farhadi, 2018*; *Li et al., 2022*).

The growing preference for single-stage models in recent years reflects their superior speed and practicality. As industries increasingly demand real-time defect detection, these efficient architectures have gained significant traction. Their ability to perform rapid and accurate analyses is critical for ensuring the performance and longevity of photovoltaic installations, where timely identification of defects is essential to maintaining operational efficiency.

## YOLOv8 algorithm overview

YOLOv8 is a state-of-the-art single-stage, anchor-free object detection algorithm, consisting of three key components: the backbone network, the neck network, and the head network.

The backbone network is responsible for extracting image features and providing the foundational feature representations required for target detection tasks. In YOLOv8, the backbone network has been significantly enhanced compared to its predecessors. It replaces all C2 modules with CSPLayer_Conv modules, introduces more skip connections, and incorporates additional split operations to improve information flow and transmission across the network.

The neck network, positioned between the backbone and head networks, plays a critical role in processing and fusing the features extracted by the backbone. This component is designed to enhance detection accuracy by combining feature maps at multiple scales, enabling the detection of objects of varying sizes. Typically, the neck network integrates structures such as the Feature Pyramid Network (FPN) and Path Aggregation Network (PAN) to achieve this multi-scale feature fusion.

The head network is tasked with generating the final detection results. YOLOv8 adopts a decoupled head structure, which separates the classification and regression branches, allowing them to operate more independently for improved performance. Additionally, it employs the Complete Intersection over Union (CIoU) metric to evaluate the differences

in the center point, width, and height of predicted bounding boxes, ensuring a more accurate representation of the target object's shape and precise localization.

In summary, YOLOv8 represents a significant advancement in single-stage object detection, with its innovative design and architectural improvements yielding superior accuracy, robustness, and efficiency in detecting objects of various sizes and shapes.

# IMPROVED YOLOV8 ALGORITHM

In the operation and maintenance of photovoltaic power plants, infrared sensing devices are commonly used to capture images of photovoltaic panels for defect localization. However, this approach is often hindered by challenges such as complex backgrounds, noise interference, and material variations. Traditional target recognition networks struggle to extract sufficient key information, resulting in poor defect feature recognition, suboptimal detection accuracy, weak model generalization, and higher rates of false positives and missed detections.

This article presents a series of structural enhancements to the YOLOv8 architecture, with the improved network architecture illustrated in Fig. 1. The proposed modifications encompass several key innovations: First, we introduce GhostConv as a replacement for the conventional convolution layer in the initial stage of YOLOv8's backbone network, effectively reducing model parameters while enhancing detection precision. Second, we integrate the BoTNet architecture into the Spatial Pyramid Pooling-Fast (SPPF) module, enabling more effective capture of feature correlations and importance weights, thereby improving overall model performance. Third, we propose and implement a novel multi-channel squeeze-excitation network (MCSENet) within the neck network, significantly enhancing feature extraction capabilities and improving representational capacity. Finally, we replace the original Complete Intersection over Union (CIoU) loss function with Focaler-CIoU, addressing the challenges associated with varying detection difficulties across different samples while accelerating model convergence and improving detection performance.

## Multi-channel squeeze-excitation network

The multi-channel squeeze-excitation network (MCSENet) enhances channel representations by employing a multi-channel squeeze-excitation mechanism, thereby improving its expressive power (*Hu, Shen & Sun, 2018*). However, with only a single squeeze-excitation feature extraction branch, its feature extraction capability remains limited and requires further enhancement (*Targ, Almeida & Lyman, 2016*).

The squeeze module compresses the input feature map *via* a fully connected layer, reducing its dimensionality before performing channel-wise multiplication operations. The excitation module applies a gating mechanism with a sigmoid activation function. This process consists of two fully connected layers: a dimension reduction layer followed by a rectified linear unit (ReLU) activation function, and a dimension restoration (increase) layer. This modular design not only limits model complexity but also facilitates generalization. The final output is produced by performing channel-wise multiplication

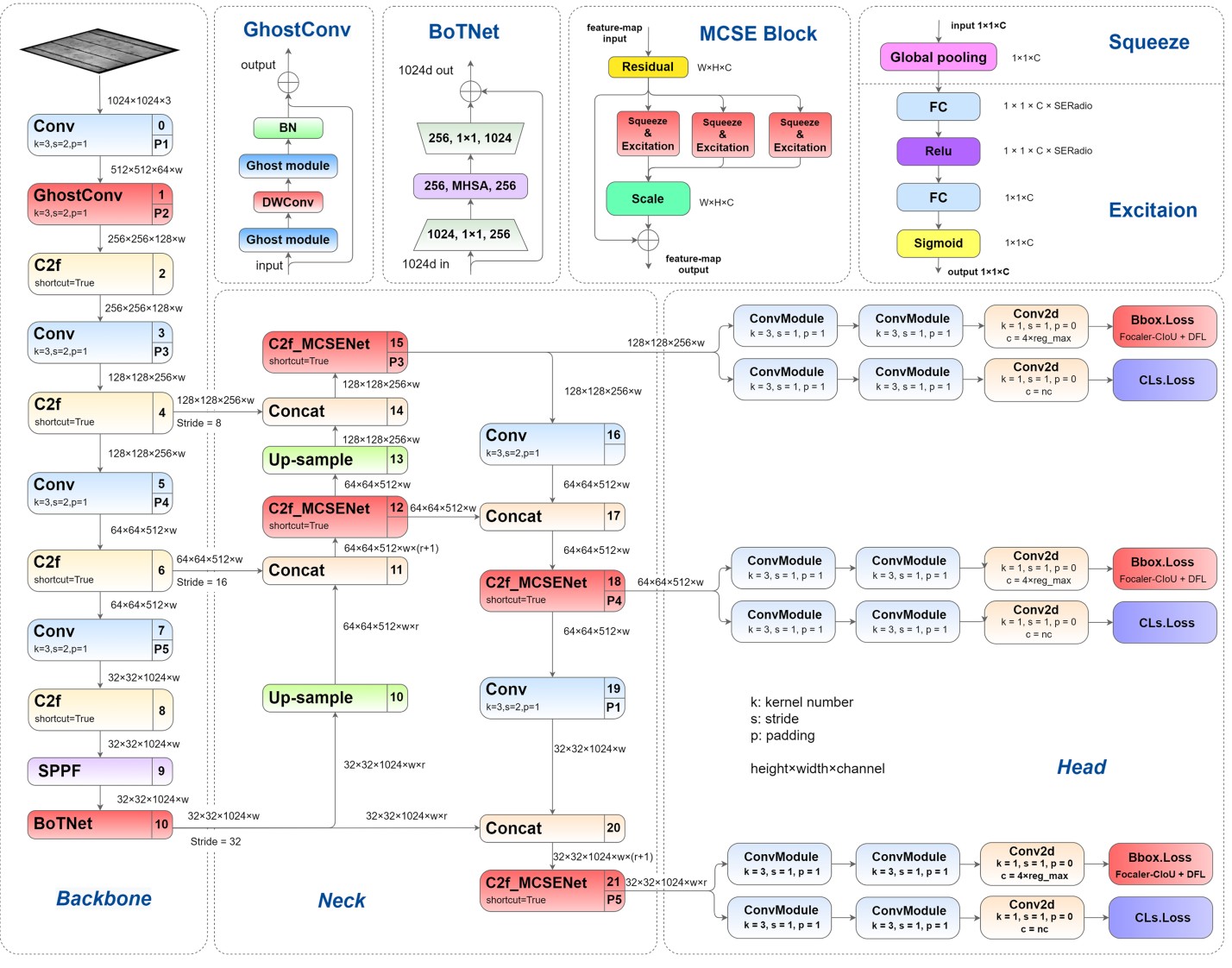

**Figure 1 Improved YOLOv8 network structure.**

between the input feature map and the computed channel weights. The detailed structure of the MCSE module is illustrated in Fig. 2.

In Fig. 2, $F_{tr}(\ )$ represents a conventional convolution operation. $F_{squeeze}(\ )$ denotes the squeeze operation, while $F_{excitation}(\ )$ signifies the excitation operation. $F_{scale}()$ is applied to rescale the output to align with the original shape. The calculation formula for the squeeze-excitation operation is expressed in Eq. (1).

$$Z_c = F_{squeeze}(U) = \sum_{i=1}^{H} \sum_{j=1}^{W} U(i,j) \tag{1}$$

where $U$ represents the feature map after conventional convolution, $H$ and $W$ denote the height and width of the feature map, respectively, and $z$ is the channel statistics obtained

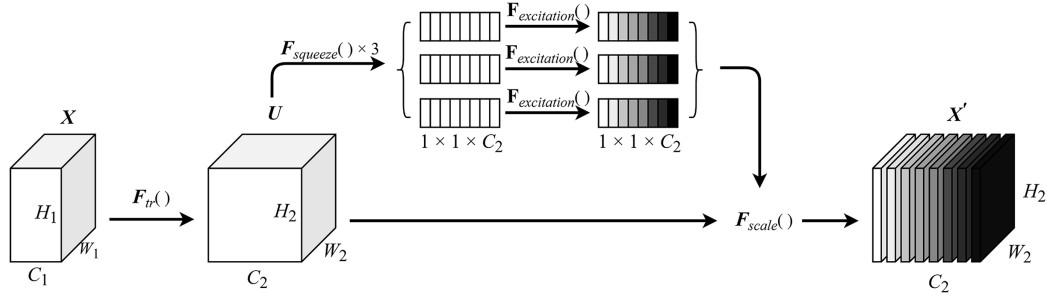

**Figure 2 MCSE module structure.**

through global average pooling. The calculation formula for the excitation operation is shown in Eq. (2).

$$s = F_{excitation}(z, \varphi) = \alpha(\varphi_2 \beta(W\varphi_1 z)). \tag{2}$$

$\alpha$ represents sigmoid activation function, $\beta$ represents ReLU function, $\varphi_1, \varphi_2 \in R^{\frac{C^2}{r}}$, the weight matrix representing two fully connected layers is used to parameterize the gating mechanism to limit model complexity and facilitate generalization ability by forming a dimension reduction layer and a dimension increase layer. The calculation of the feature map $u_c$ is shown in Eq. (3).

$$u_c = v_c * X = \sum_{s=1}^{C'} v_c^s * X^s. \tag{3}$$

Let $v_c$ denote the learned set of filter kernels, where $v_c$ is the parameter of the $c$th filter used to map the input $X$ to the feature map $U, X, U \in R^{H_1 \times W_1 \times C_1}$. $s_c$ represents the channel weight, which is used to channel weight the feature map $u_c$. $X'$ is the corresponding channel product between feature map $u_c$ and channel weight $s_c$, as shown in Eq. (4).

$$X' = F_{scale}(u_c, s_c) = s_c u_c = s_c \sum_{s=1}^{C'} v_c^s * X^s \tag{4}$$

where the inner brackets after $fc$ indicate the output dimension of the two fully connected layers in the squeeze-excitation module.

In MCSENet, the feature extraction capability can be enhanced by increasing the number of feature extraction branches. When the output of the squeeze-excitation fully connected layer is concatenated along the channel dimension, an excessive number of channels in the squeeze-excitation fully connected layer can diminish the influence of the original fully connected layer on the concatenated result. This means that when the feature map after the squeeze-excitation operation is merged with the original feature map, the features of the original feature map are diluted, and the features extracted by squeeze-excitation are overly emphasized. This phenomenon is analogous to the network degrading to a version without a residual structure, thereby affecting the network's detection accuracy.

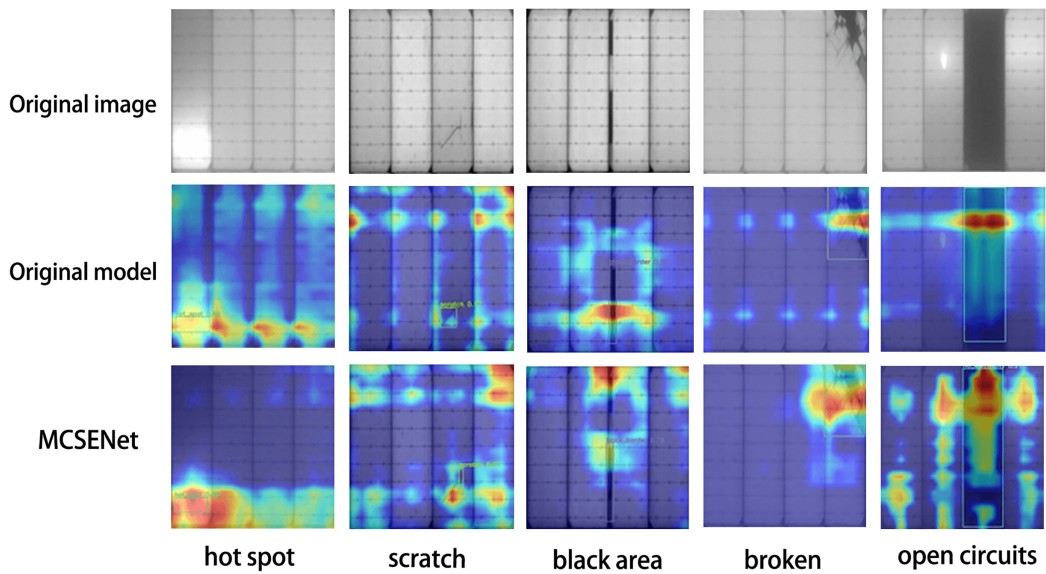

**Figure 3 MCSENet heat map was introduced for comparison.**

To validate the focus of MCSENet on defect features, five images were selected from the PV-Multi-Defect-main dataset to visualize the feature extraction performance of the YOLOv8 baseline model and the model incorporating MCSENet, as illustrated in Fig. 3. The heat maps show that after the introduction of MCSENet, the network's attention to the defect location becomes more precise, and the interference from background factors is reduced, positively impacting the defect target localization during photovoltaic panel defect detection.

## GhostConv

In the convolution process, it is crucial to consider the redundancy of feature maps within the model structure. The robust feature extraction capability of convolutional neural networks (CNNs) is positively correlated with these similar feature maps, often referred to as Ghost feature maps (*Lau, Po & Rehman, 2024*). Rather than deliberately avoiding the generation of such Ghost feature maps, GhostConv aims to leverage simple linear operations to produce an increased number of these features in order to enhance the model's feature extraction ability.

The GhostConv convolution module (*Han et al., 2020*) addresses the issue of redundancy in feature maps by proposing a Ghost module structure that generates a substantial quantity of feature maps with minimal computational cost. This module creates mappings for intrinsic features through conventional convolution and subsequently amplifies both the number of features and channels *via* depth-wise convolution (DC) (*Chollet, 2017*), effectively replacing pointwise convolution while utilizing depthwise convolutions for spatial information processing. Consequently, Ghost convolution not only improves detection performance but also reduces computational requirements.

A convolution operation generates an output feature map by multiplying the input data with a convolution filter and adding a bias term. This operation involves a large number of floating-point operations, and usually needs to deal with a large number of parameters and computational complexity, especially when the output feature map of the convolution layer contains a lot of redundant information. The computational complexity can be calculated by $n \cdot h' \cdot w' \cdot c \cdot k^2$, where $n$ is the number of feature maps, $h'$ and $w'$ are the height and width of the output data, $c$ is the number of input channels, and $k$ is the size of the convolution kernel. This computational complexity is typically very high, since typically both $n$ and $c$ are large. The operation of an arbitrary convolution layer to generate $n$ feature maps can be expressed as Eq. (5).

$$Y = X*f + b. \tag{5}$$

$X$ represents the input data, * for the convolution operation, $b$ is the amount of bias, $f$ represents a convolution filter in a convolution layer, $Y$ is the input feature map. Equation (6) describes the process of generating the intrinsic feature map in the Ghost module.

$$Y' = X*f'. \tag{6}$$

$m$ intrinsic feature maps, $Y' \in R^{h' \times w' \times f'}$ is generated by regular convolutions. Where $f' \in R^{c \times k \times k \times m}$ is filter, $m \leq n$. These intrinsic feature maps can be viewed as smaller feature maps produced by a conventional convolution filter. Next, to further obtain the desired $n$ feature maps, for each intrinsic feature in $Y'$, a series of low-computation linear operations are performed to generate $s$ "ghost" features. The linear operation with a low calculation amount can be expressed as Eq. (7):

$$y_{ij} = \Phi_{i,j}(y'_i), \quad \forall i = 1, \ldots, m; \quad j = 1, \ldots, \tag{7}$$

where $y'_i$ is the $i$-th intrinsic feature map of $Y'$, $\Phi_{i,j}$ represents the $j$th linear operation that generates the $j$-th ghosting feature map $y_{ij}$. The regular convolution operation and the Ghost convolution operation are shown in Fig. 4.

Suppose that the number of generated channels after convoluting input with the kernel of $c'$ groups $k \times k$ in the original convolution operation is $c'$ and the size of the output dimension is $h' \times w'$. In the ghost model, $c_{mid}$ groups of $k \times k$ kernels are convolved with the input to generate the eigenmap of $c_{mid} \times h' \times w'$. After that, the eigenmap is linearly transformed by $\Phi$ to generate the ghost map, and the eigenmap and the ghost feature map are taken as the output.

## Bottleneck transformer network

Bottleneck Transformer Network (BoTNet) (*Srinivas et al., 2021*) is a Transformer-based backbone network architecture designed to address the challenge of long-distance dependency modeling. By incorporating the multi-head self-attention (MHSA) mechanism, BoTNet effectively captures feature correlations and their importance, thereby enhancing model performance. This design allows BoTNet to outperform traditional

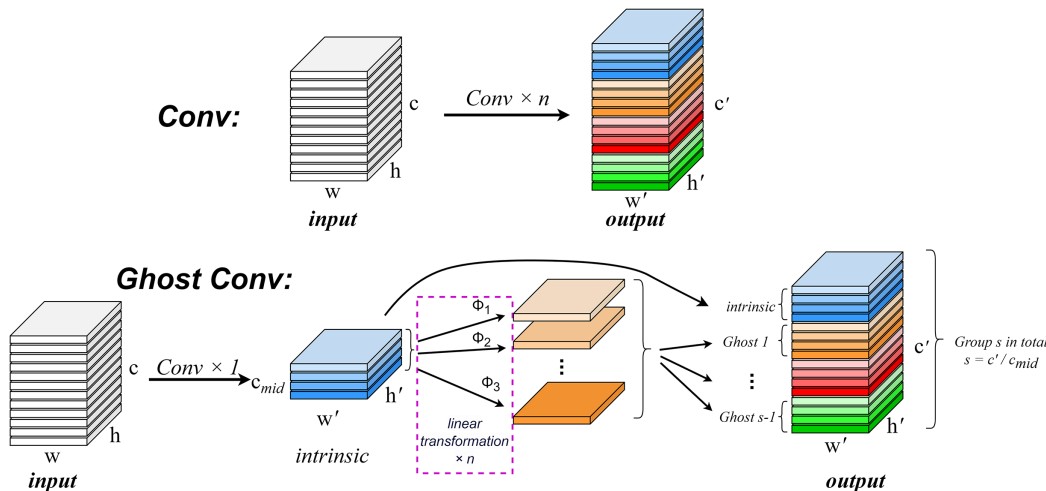

**Figure 4 Comparison of normal convolution operation (top) and ghost convolution operation (bottom).**

architectures when replacing the ResNet backbone, while simultaneously reducing the number of model parameters and improving both efficiency and accuracy. Unlike the traditional Transformer architecture (*Vaswani et al., 2017*), BoTNet simplifies model hierarchy by replacing spatial convolutions in the last three bottleneck blocks with the MHSA mechanism. MHSA offers distinct advantages over conventional sequence modeling methods, including its ability to handle long-range dependencies and support parallel computation, making it highly efficient for vision tasks.

The structure of the BoTNet module and the workflow of the MHSA layer are illustrated in Fig. 5. $q, k, r$ denote the query code, key code, and position code, respectively. The input size of MHSA is $H \times W \times d$, respectively represent the height and width of the input feature matrix and the dimension of a single token, and the number of tokens is $H \times W$. The first is to initialize two learnable parameter vectors $R_h$ and $R_w$, represent the position codes at different positions of height and width. Respectively, and added by the broadcast mechanism, The encoding of position $(i, j)$ is the sum of two $d$-dimensional vectors $R_{hi}$ and $R_{wj}$, and simplified the numbers of encoding from $H \times W \times d$ to $(H + W) \times d$. The position code is multiplied with the query matrix, and then multiplied with the query and key matrices. The result is summed up, and finally normalized by softmax to obtain the final attention.

## Focaler-CIoU

Focaler-CIoU (*Zhang & Zhang, 2024*) is based on CIoU (*Zheng et al., 2020*) and uses the linear margin mapping method to reconstruct the Intersection over Union (IoU) loss, which helps to improve the marginal regression. The loss function used in YOLOv8 model is CIoU loss function. By adding a new shape loss term in Distance Intersection over Union (DioU), CIoU further considers the shape similarity between

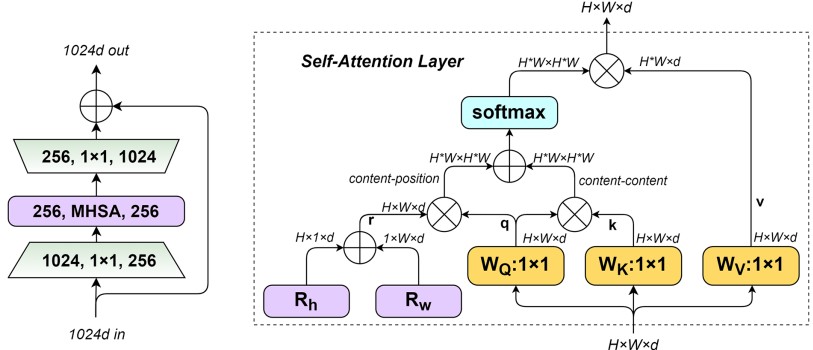

**Figure 5 BoTNet module structure (left), MHSA layers used in BoTNet blocks (right).**

anchor frames to reduce the difference of aspect ratio between anchor frames. It is defined as follows:

$$CIoU = IoU - \frac{\rho^2(b, b^{gt})}{c^2} - \alpha v \tag{8}$$

$$IoU = \frac{|B \cap B^{gt}|}{|B \cup B^{gt}|} \tag{9}$$

$$\alpha = \frac{v}{(1 - IoU) + v} \tag{10}$$

$$v = \frac{4}{\pi^2} \left( arctan \frac{w^{gt}}{h^{gt}} - arctan \frac{w}{h} \right)^2 \tag{11}$$

$$Loss_{CIoU} = 1 - CIoU = 1 - IoU + \frac{\rho^2(b, b^{gt})}{c^2} + \alpha v \tag{12}$$

In the formula, $IoU$ is the intersection and union ratio, $B$ and $B^{gt}$ represent the predicted and ground true (GT) boxes, respectively, $b$ and $b^{gt}$ represents the center points of two rectangular boxes, $\rho$ represents the Euclidean distance between two rectangular boxes, $c$ represents the diagonal distance of the enclosed area of the two rectangular boxes, $v$ used to measure the consistency of the relative proportions of two rectangular boxes, $\alpha$ is the weight coefficient, $w^{gt}$, $h^{gt}$ is the width and height of the GT boxes. $w$, $h$ is the width and height of the anchor frame.

The advantage of CIoU is that it takes the shape into account and introduces a correction factor $v$, which makes it easier for the model to capture the exact shape of the target. Due to the introduction of diagonal distance, the CIOU loss function helps to improve the accuracy of the target detection model in localization. However, for the problem of sample imbalance, CIoU can cause huge IoU change loss for samples that are difficult to distinguish, such as small targets, which can easily cause adverse effects on the regression effect of the bounding box, and the judgment effect is also poor for targets with a large ratio of length to width. In the defect detection of photovoltaic panels, it is difficult to

detect small defects such as broken grids and scratches, so YOLOv8, which uses CIoU as the loss function of bounding box, has poor accuracy and generalization in the detection of the above small targets.

In order to improve the calculation method of loss function to improve the accuracy of localization, the Focaler-CIoU loss function is proposed. The Focaler-IoU is applied to CIoU, which can improve the detection effect by focusing on different regression samples in different detection tasks. IoU loss is reconstructed by the method of fetching mapping, and the calculation method of Focaler IoU loss is as shown in Eq. (13).

$$IoU^{focaler} = \begin{cases} 0, & IoU < d \\ \dfrac{IoU - d}{u - d}, & d \ll IoU \ll u \\ 1, & IoU > d. \end{cases} \tag{13}$$

$IoU^{focaler}$ is Focaler-IoU after reconstruction, $IoU$ is original $IoU$ Value, $[d, u] \in [0, 1]$. By regulation $d$ and $u$ are value of is such that $IoU^{focaler}$ focus on different regression samples. The loss is defined as shown in Eq. (14):

$$L_{Focaler-IoU} = 1 - IoU^{focaler}. \tag{14}$$

Applying Focaler-IoU to the existing IOU-based bounding box regression loss function, the definition of $L_{Focaler-CIoU}$ is given in Eq. (15).

$$L_{Focaler-CIoU} = L_{CIoU} + IoU - IoU^{Focaler} = 1 + \frac{\rho^2(b, b^{gt})}{c^2} + \alpha v - IoU^{Focaler}. \tag{15}$$

## IMPLEMENTATION DETAILS AND EVALUATION METRICS

In order to evaluate the scientific validity and effectiveness of the proposed method, a comprehensive set of experimental procedures was meticulously arranged and designed. Firstly, we introduce the dataset utilized in the experiment, along with details regarding the experimental environment and parameters. Subsequently, experiments were conducted to assess how varying the number of channels in the proposed MCSENet affects defect feature extraction.

Secondly, we discuss the impact of different layers of GhostConv convolution on defect feature detection performance by testing various configurations involving both layer counts and numbers of GhostConv convolutions. Thirdly, we compare the influence of IoU loss functions on model accuracy to validate the efficacy of Focaler-CIoU in enhancing model precision. This is followed by a horizontal comparison with other improvement methods to establish that our enhanced approach offers significant advancements.

Finally, ablation studies were performed on our proposed method and compared against other models to ensure that each step taken in this enhancement process contributes positively towards improving detection accuracy within our model.

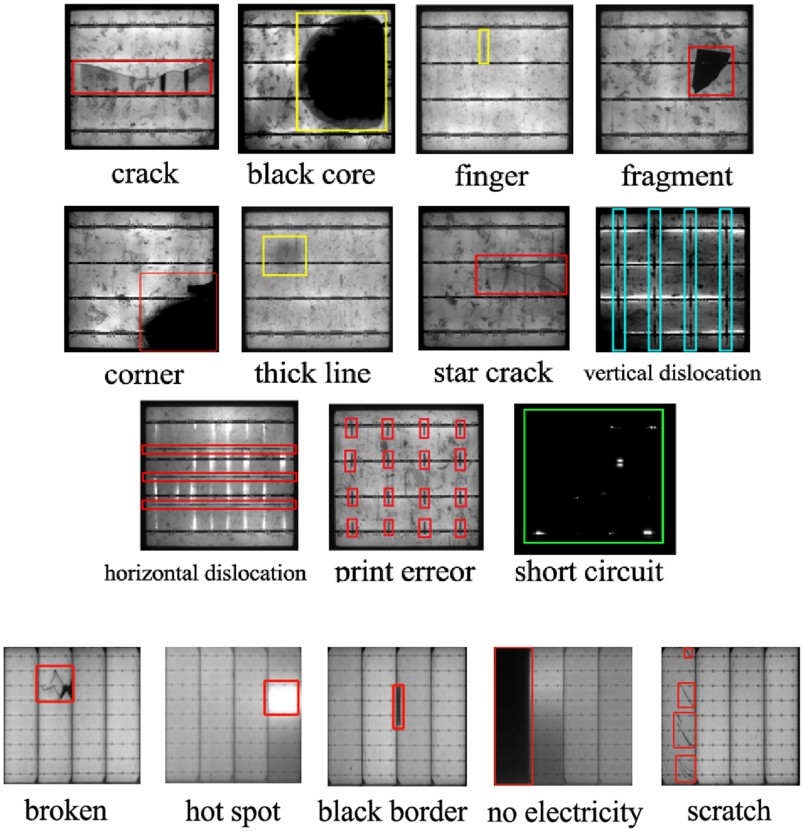

**Figure 6 Sample PVEL-AD (top), PV-Multi-Defect-main (bottom) defect type picture.**

## Dataset introduction

This article utilizes two datasets: the PV-Multi-Defect-main dataset (*Li, Wang & Zhang, 2023*). The PV-Multi-Defect-main dataset contains 1,106 images of photovoltaic panel defects, categorized into five types: 2,079 hot spots, 1,367 scratches, 256 open circuits, 181 black areas, and 98 broken areas. The image size is 600 pix * 600 pix. Scratches and hot spots are relatively small and difficult to detect. Open circuits and black areas have large defect areas and are easy to detect.

The PVEL-AD dataset (*Su, Zhou & Chen, 2022*), the world's largest dataset for photovoltaic panel defect anomaly detection, was jointly released by Hebei University of Technology and Beijing University of Aerospace and Technology (PVEL-AD dataset: https://github.com/binyisu/PVEL-AD). It comprises 4,997 images of photovoltaic panel defects, each with a resolution of 1,024 pix * 1,024 pix, and includes 11 defect types: 1,260 cracks, 2,958 fingers, 1,028 black cores, 981 thick lines, 135 star cracks, 128 corners, 112 fragments, 798 horizontal dislocations, 137 vertical dislocations, 32 printing errors, and 492 short circuits. There are all kinds of defects in the PVEL-AD dataset, among which the defects of the fingers type belong to the category of small targets and are the most difficult

to detect. Sample images of defect types from both the PV-Multi-Defect-main and PVEL-AD datasets are shown in Fig. 6.

The PV-Multi-Defect-main dataset and PVEL-AD dataset cover the main types of defects that can be captured through electroluminescence imaging of photovoltaic panels, including defects caused during the production, installation, and operation of the panels.

In this study, the data augmentation techniques employed include spatial geometric transformations, such as rotation, flipping, and affine transformations. Specifically, rotation was applied to randomly rotate images within a predefined angle range to improve the model's robustness to different object orientations. Flipping was used to mirror the images horizontally or vertically, helping the model generalize better by exposing it to variations in object positioning. Affine transformations were applied to modify the geometric properties of the images, such as scaling, shearing, and translation, allowing the model to learn spatial invariance and improve its performance on diverse image perspectives. These augmentation techniques collectively help enhance the model's ability to generalize to unseen data, reduce overfitting, and improve overall detection accuracy. The PV-Multi-Defect-main dataset was expanded to 4,454 images. We compared the detection performance of a model trained on the enhanced dataset with a model trained on the original dataset. In the PVEL-AD dataset, fragment and corner defect types are underrepresented, resulting in significant class imbalances that can adversely affect model performance. Therefore, only fragmentation and lack of sleep are enhanced.

In this study, the dataset was divided into training, testing, and validation sets in a ratio of 7.5:1:1.5. This division was primarily chosen to ensure that the validation set plays a crucial role in adjusting the model's hyperparameters and preventing overfitting. The validation set is especially critical when the dataset size is relatively small, requiring an increased proportion.

For images containing multiple defects, either of the same type or different types, such cases are present in both datasets used in this study. All defects in the images were annotated during the data labeling process. This approach aligns with current methodologies for handling multi-defect datasets.

## Environment and training parameters

The experimental environment and training parameters is shown in Table 1.

The training parameters in Table 1 are typically the default settings for YOLOv8. Comparative experiments have verified that these parameters are the most suitable for this experiment. For instance, stochastic gradient descent (SGD) uses the data of only one sample to compute the gradient, resulting in faster convergence. Additionally, by introducing random noise during the training process, SGD can prevent the model from overfitting the training data, thereby improving its performance on new data.

YOLOv8 automatically stops training to prevent overfitting and resource wastage when no improvement is observed after 50 iterations, indicating that the model has reached convergence. Consequently, the actual number of iterations is often less than the number

**Table 1 Experimental environment and training parameter setting.**

| Environment | Parameter |
| --- | --- |
| Operating system | Ubuntu 18.04.6 LTS (64 bit) |
| CPU | Intel Xeon(R) Gold 6240 CPU @ 2.60 GHz * 4 |
| GPU | NVIDIA Tesla T4 16 GB |
| Memory | 8G*2 |
| Python | 3.8.18 |
| CUDA | 10.1 |
| Pytorch | 1.9.0 |
| Numpy | 1.24.3 |
| Batch size | 8 |
| Initial learning rate | 0.01 |
| Minimum learning rate | 0.0001 |
| Momentum | 0.937 |
| Weight decay | 0.0005 |
| Optimizer | SGD |
| Imgsz | 600 or 1,024 |
| Close mosaic | 10 |
| Amp | False |

set in the hyperparameters to ensure complete convergence of the loss and mAP during training. For the PV-Multi-Defect-main dataset, convergence is typically achieved after about 300 iterations without data enhancement. However, with data enhancement, the model converges more slowly, generally requiring about 1,200 iterations. For the PVEL-AD dataset, which features high resolution and a large number of images, convergence is usually reached after 600 iterations.

In analyzing the experimental results, the COCO evaluation metrics are used to assess model performance. These include mean average precision (mAP), precision (P), recall (R), giga floating-point operations per second (GFLOPs), the number of parameters, and frames per second (FPS) . Observe the accuracy of the model on the training and validation sets. If the accuracy of the training set is much higher than that of the validation set, it indicates that overfitting has occurred.

The convolution, attention mechanisms, and IoU code used in these experiments are documented in the referenced articles. When using these mechanisms to reproduce the experiments, adhere to the following principles: (1) Ensure that the software, hardware, and hyperparameters used in the experiment are identical. (2) Modify the original YOLOv8n model as needed. (3) If the developer of the convolution or attention mechanism has recommended a YAML file structure, modify it accordingly; otherwise, replace all applicable convolution or attention mechanisms in the YAML file. (4) Use the same dataset and its division.

**Table 2 Comparison of detection effect of number of squeeze-excitation channels.**

| Numbers of channels | mAP50 | mAP50-95 | Precision | Recall | Parameter |
|---|---|---|---|---|---|
| 1 | 0.818 | 0.505 | 0.790 | 0.758 | 3,015,023 |
| 2 | 0.832 | 0.513 | 0.774 | 0.805 | 3,018,223 |
| 3 | 0.845 | 0.532 | 0.793 | 0.819 | 3,021,423 |
| 4 | 0.843 | 0.525 | 0.794 | 0.801 | 3,024,623 |
| 5 | 0.804 | 0.495 | 0.782 | 0.791 | 3,027,823 |

# EXPERIMENTAL RESULTS AND DISCUSSION

## Effect comparison of MCSENet channel number on defect feature detection

MCSENet features a greater number of squeeze-excitation channels compared to traditional SENet, thereby enhancing its feature extraction capabilities. To ascertain the optimal quantity of squeeze-excitation channels, experiments were undertaken using the unenhanced PV-Multi-Defect-main dataset, with YOLOv8n serving as the foundational detection network. The performance metrics for detecting defects in photovoltaic panels encompassed mAP50(%), mAP50-95(%), precision, recall, and the parameter count. The experimental findings are summarized in Table 2.

An analysis of the results presented in Table 2 reveals that augmenting the number of branches in the squeeze-excitation (SE) channels bolsters the model's feature extraction prowess, adopting a structure akin to that of a residual network. However, incorporating an excessive number of SE channels can result in the dilution of the original feature map when fused with the map post-squeeze-excitation. This imbalance leads the model to overly prioritize features extracted *via* squeeze-excitation, mirroring the degradation observed in models devoid of a residual structure, thereby detracting from detection accuracy. While the proliferation of SE channel branches can indeed enhance the model's feature extraction capabilities due to its structural resemblance to a residual network, an overly abundant inclusion of compression-excitation channels can lead to the dilution of the original feature map during the merging process post-SE operation in scaling. This heightened emphasis on SE-extracted features may prompt the network to degrade into one with fewer residual blocks, ultimately undermining the network's detection accuracy. Consequently, an excessive number of SE channels can diminish the model's detection performance.

Consequently, an excessive number of squeeze-excitation channels can reduce the model's detection performance. The optimal detection performance is achieved with 3 or 4 channels. The same conclusion was reached when using the PVEL-AD dataset.

## Comparison of detection effect of layer number of Ghost convolution on defect feature

In the PV-Multi-Defect-Main dataset, scratches and cracks on PV panels appear stripe-like. During the feature extraction process in the YOLOv8 backbone network,

**Table 3 Comparison of effects of GhostConv in different layers of the backbone network.**

| GhostConv is on | mAP50 | mAP50-95 | Precision | Recall | Parameter |
|---|---|---|---|---|---|
| 1st floor | 0.849 | 0.541 | 0.807 | 0.752 | 3,004,719 |
| 3rd floor | 0.807 | 0.498 | 0.794 | 0.708 | 2,998,207 |
| 5th floor | 0.793 | 0.484 | 0.734 | 0.692 | 2,971,359 |
| 7th floor | 0.781 | 0.493 | 0.683 | 0.751 | 2,862,367 |
| 1st, 3rd floors | 0.842 | 0.532 | 0.812 | 0.819 | 3,001,487 |
| 1st, 5th floors | 0.836 | 0.518 | 0.802 | 0.738 | 2,974,639 |
| 1st, 7th floors | 0.833 | 0.528 | 0.744 | 0.817 | 2,860,463 |
| 1st, 3rd, 5th floors | 0.831 | 0.530 | 0.810 | 0.781 | 2,961,039 |
| 1st, 5th, 7th floors | 0.826 | 0.521 | 0.802 | 0.751 | 2,825,199 |
| 1st, 3rd, 7th floors | 0.792 | 0.483 | 0.712 | 0.698 | 2,852,047 |
| 1st, 3rd, 5th, 7th floors | 0.842 | 0.545 | 0.767 | 0.727 | 2,821,967 |

shallow layers typically extract detailed features. However, traditional convolutional layers are less effective at extracting defect features and have high computational overhead. To address this, GhostConv is employed in the shallow layers of YOLOv8 instead of traditional convolutions. This substitution enhances the detection accuracy for scratch and crack types while reducing computational costs. The output of these shallow layers is then utilized by deeper network layers, further improving overall detection accuracy.

To evaluate the impact of GhostConv on model accuracy and efficiency, experiments were conducted comparing the effects of replacing Conv with GhostConv at different network layers. The results of these comparisons are displayed in Table 3.

The feature extraction ability of convolution is positively correlated with the Ghost feature map. GhostConv generates the Ghost feature map using linear operations with low computational complexity, achieving both lower computational complexity and improved results. By incorporating GhostConv into the shallow layers of the backbone network, the generated Ghost feature map can be passed to deeper layers, enhancing the overall feature extraction capability of the network. In the deeper layers of the backbone network, its detection performance may decrease, and it is not as good as in the shallow layers. Thus, the shallower the layer, the better the feature extraction effect. The same conclusion was reached when using the PVEL-AD dataset. Due to the distortion caused by low-computation linear operations, when Ghost feature maps are further passed through additional low-computation linear operations in the GhostConv convolution, this distortion effect is amplified, negatively impacting the feature extraction capability. Therefore, an excessive number of GhostConv layers can reduce the feature extraction ability of the backbone network. Additionally, GhostConv layers located in deeper layers of the network may further exacerbate the distortion during the upsampling process in the neck network, affecting the features of large objects and ultimately leading to a decrease in detection accuracy.

**Table 4  Test results of different IoU loss functions.**

| Loss function | mAP50 | mAP50-95 | Precision | Recall |
|---|---|---|---|---|
| EIoU | 0.847 | 0.532 | 0.823 | 0.824 |
| CIoU | 0.813 | 0.506 | 0.777 | 0.712 |
| WIoU | 0.827 | 0.515 | 0.809 | 0.785 |
| SIoU | 0.813 | 0.509 | 0.846 | 0.736 |
| InnerIoU | 0.790 | 0.482 | 0.734 | 0.727 |
| FocalerIoU | 0.836 | 0.507 | 0.793 | 0.800 |
| Focaler-SIoU | 0.840 | 0.543 | 0.813 | 0.786 |
| Focaler-EIoU | 0.835 | 0.529 | 0.798 | 0.771 |
| Focaler-WIoU | 0.730 | 0.328 | 0.528 | 0.737 |
| Focaler-CIoU | 0.857 | 0.539 | 0.795 | 0.835 |

## IoU selection

The loss function plays a crucial role in photovoltaic panel defect target detection. In YOLOv8, the loss function consists of category classification loss and frame regression loss, the latter of which includes distribution focal loss (DFL) and IoU loss. By minimizing the loss function, the model learns to accurately locate targets, predict their existence and categories, and ultimately improve target detection accuracy. To investigate the impact of different IoU loss functions on model performance, a comparative test was conducted using CIoU, EIoU, WIoU, InnerIoU, FocalerIoU, and their combinations on the PV-Multi-Defect-main dataset. The experimental results are presented in Table 4.

Table 4 shows that changing the IoU loss function affects the model's detection performance, with minimal impact on the number of model parameters. Using Focaler-CIoU as the loss function enhances the model's prediction accuracy to varying degrees compared to other IoU loss functions. Thus, Focaler-CIoU is selected as the model's bounding box regression loss function to achieve better photovoltaic panel defect detection.

## Comparison of other improved model

To verify the effectiveness of MCSENet and various improvements proposed in this article, and to ensure the advancement of the improved model, we compared the detection performance of enhanced modules based on YOLOv8. These comparisons include improvements in the convolution kernel and attention mechanism. The models compared include GoldYOLO (Wang et al., 2024), deep & cross network version 2 (DCNv2) (Wang et al., 2021), deep & cross network version 3 (DCNv3), weighted bi-directional feature pyramid network (BiFPN) (Tan, Pang & Le, 2020), Omni-dimensional Dynamic Convolution (ODConv) (Li, Zhou & Yao, 2022), Expectation-maximization attention (EMA) (Li et al., 2019), and Large separable kernel attention (LSKA) (Lau, Po & Rehman, 2024), among others. The PV-Multi-Defect-main dataset, without enhancement processing, was used for these experiments, with an image resolution of 600 × 600 pixels. Hyperparameters and training parameters were consistent with those outlined in the section

**Table 5 Effect comparison of different improvement methods based on YOLOv8.**

| Improved model | mAP50 | mAP50-95 | Precision | Recall |
|---|---|---|---|---|
| YOLOv8+GoldYOLO | 0.848 | 0.533 | 0.756 | 0.870 |
| YOLOv8+C2f_DCNv2 | 0.836 | 0.513 | 0.731 | 0.766 |
| YOLOv8+C2f_DCNv3 | 0.834 | 0.529 | 0.817 | 0.767 |
| YOLOv8+ODconv | 0.827 | 0.515 | 0.664 | 0.842 |
| YOLOv8+BiFPN | 0.821 | 0.503 | 0.731 | 0.812 |
| YOLOv8+CA | 0.817 | 0.456 | 0.725 | 0.811 |
| YOLOv8+ECA | 0.819 | 0.513 | 0.767 | 0.746 |
| YOLOv8+MLCA | 0.834 | 0.514 | 0.791 | 0.799 |
| YOLOv8+CBAM | 0.819 | 0.469 | 0.724 | 0.805 |
| YOLOv8+EMA | 0.827 | 0.528 | 0.758 | 0.830 |
| YOLOv8+LSKA | 0.841 | 0.527 | 0.816 | 0.800 |
| YOLOv8+Small target detection head | 0.829 | 0.530 | 0.756 | 0.790 |
| Ours | 0.861 | 0.551 | 0.813 | 0.816 |

"Experimental Environment and Training Parameters." The evaluation metrics for photovoltaic panel defect detection included mAP50(%), mAP50-95(%), precision, and recall. The results, presented in Table 5, demonstrate the effectiveness of the model improvements.

## Ablation experiment

In order to verify the validity of the defect detection method for photovoltaic panels proposed in this article and explore the impact of improvements in MCSENet, GhostConv, BoTNet and Focaler-CIoU on the model detection results, this article conducts tests on PV-Multi-Defect-main dataset and PVEL-AD dataset. The baseline model used is YOLOv8, the ablation experiments were carried out on the dataset for each proposed improved module. The other experimental environments and parameters are completely the same, only modifying and exploring the impact of modules or components on model performance.

### Ablation experiments on the PV-Multi-Defect-main dataset

Among all combinations from the ablation experiments on the PV-Multi-Defect-main dataset, the best results were achieved using two or three improvement methods, excluding data enhancement. Figure 7 shows a comparison of mAP50 values, mAP50-95 values, recall rate, and loss in the YOLOv8n model's experimental results. Results on the PV-Multi-Defect-main dataset are detailed in Table 6.

Analyzing these experimental results, it is clear that the methods proposed in this article enhance the accuracy of detecting defects in photovoltaic panels. Among the single improvement methods, apart from data enhancement, the GhostConv and MCSENet schemes performed particularly well. GhostConv increases the number of Ghost pairs through a linear operation with low computational complexity, making the model lighter

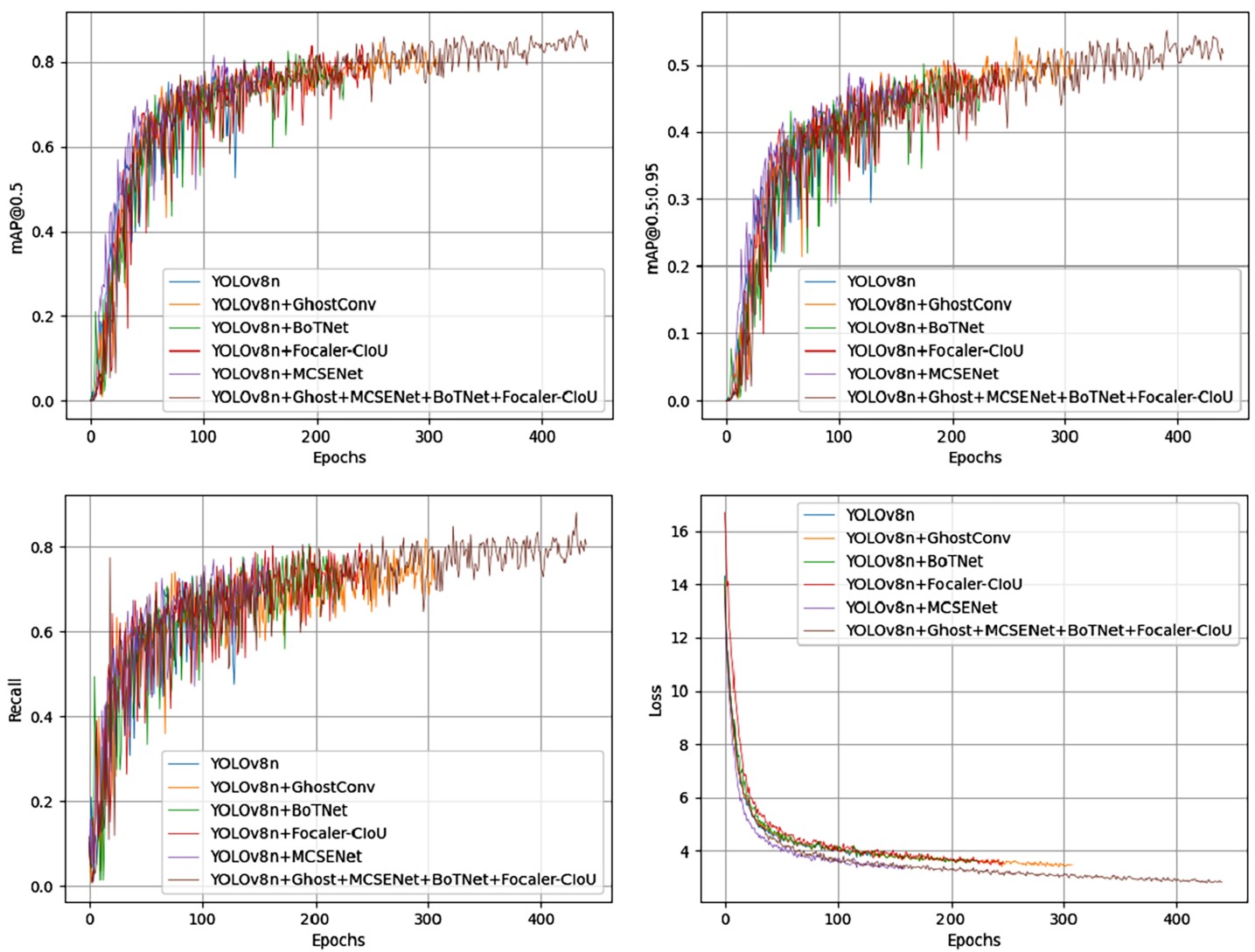

**Figure 7 mAP50, mAP50-95, recall, loss comparison of model test results on the PV-Multi-Defect-main dataset.**

and reducing both the number of parameters and computational complexity. This results in a parameter reduction of 0.32 million and a decrease in GFLOPs by 0.1 billion. The mAP50, mAP50-95, precision, and recall improved by 3.6%, 3.5%, 3%, and 4%, respectively, while FPS remained consistent with the baseline model.

MCSENet boosts the extraction of defect features and suppresses background noise through squeeze-excitation channels, significantly improving detection accuracy and recall without increasing computational load. Specifically, mAP50 improved by 3.2%, mAP50-95 by 2.6%, accuracy by 1.6%, and recall by 10%. The parameters and FPS increased only slightly.

BoTNet's introduction of MHSA enhances detection accuracy by about 1 to 2 percentage points, albeit with increased computational requirements. The Focaler-CIoU

**Table 6 Comparison of the ablation results of PV-Multi-Defect-main dataset.**

| GhostConv | BoTNet | MCSENet | Focaler-CIoU | Data enhance | mAP50 | mAP50-95 | Precision | Recall | Parameter | FLOPs (G) | FPS |
|---|---|---|---|---|---|---|---|---|---|---|---|
| | | | | | 0.813 | 0.506 | 0.777 | 0.712 | 3,006,623 | 8.1 | 263 |
| √ | | | | | 0.849 | 0.541 | 0.807 | 0.752 | 2,998,207 | 8.0 | 270 |
| | √ | | | | 0.828 | 0.537 | 0.795 | 0.748 | 3,220,767 | 8.3 | 226 |
| | | √ | | | 0.845 | 0.532 | 0.793 | 0.819 | 2,872,945 | 8.1 | 277 |
| | | | √ | | 0.857 | 0.539 | 0.795 | 0.835 | 3,006623 | 8.1 | 270 |
| | | | | √ | 0.884 | 0.597 | 0.857 | 0.824 | 3,006,623 | 8.1 | 255 |
| √ | √ | √ | | | 0.838 | 0.516 | 0.803 | 0.798 | 3,228,271 | 8.2 | 239 |
| | | √ | √ | | 0.843 | 0.533 | 0.798 | 0.756 | 3,018,719 | 8.1 | 266 |
| √ | √ | | √ | | 0.835 | 0.525 | 0.804 | 0.814 | 3,218,863 | 8.3 | 236 |
| √ | √ | √ | √ | | 0.861 | 0.551 | 0.813 | 0.816 | 3,219,567 | 8.2 | 232 |
| √ | √ | √ | √ | √ | 0.926 | 0.660 | 0.907 | 0.875 | 3,219,567 | 8.3 | 236 |

loss function reconstructs the original IoU loss *via* linear interval mapping, effectively addressing sample imbalance and improving detection accuracy. Compared to the CIOU loss function in the baseline model, mAP50 increased by 4.5%, and recall improved by 12.3%, highlighting the challenge of sample imbalance in the detection process. Data enhancement further boosts dataset accuracy by 6 to 7 percentage points.

In the proposed model, GhostConv generates Ghost features through linear operations, MCSENet enhances defect features while suppressing background information, and BoTNet's MHSA enhances the network model's feature extraction capability. Focaler-CIoU addresses the issue of unbalanced sample targets. The integration of these four modules in YOLOv8 enhances model performance on the PV-Multi-Defect-main dataset, with a slight decrease in model parameters, GFLOPs, and FPS. mAP50 increases to 86.1%, and mAP50-95 to 55.1%. Accuracy rises to 81.3%, and recall to 81.6%. Detection results for some defect samples are shown in Fig. 8, and the P-R curves of the improved models on the PV-Multi-Defect-main dataset are displayed in Fig. 9.

### Ablation experiments on the PVEL-AD dataset

Figure 10 shows the comparison of mAP50, mAP50-95, recall rate, and loss from the ablation experiments on the PVEL-AD dataset. In the proposed model, the loss function steadily decreases and converges after about 300 epochs on the PVEL-AD dataset. The improved model demonstrates strong performance on this dataset, with the ablation experiment results presented in Table 7.

In Table 7, an image enhancement method based on an improved CycleGAN is mentioned, which primarily addresses issues such as uneven brightness and stain interference in photovoltaic panels. To prevent academic ethics concerns and ensure rigor and objectivity, this method is not described in detail here. Furthermore, the enhanced images are not displayed, as this article has not yet been published, and I cannot present results that have not been peer-reviewed. If this article is successfully published, I will contact PeerJ to include a proper citation. This CycleGAN-based image enhancement

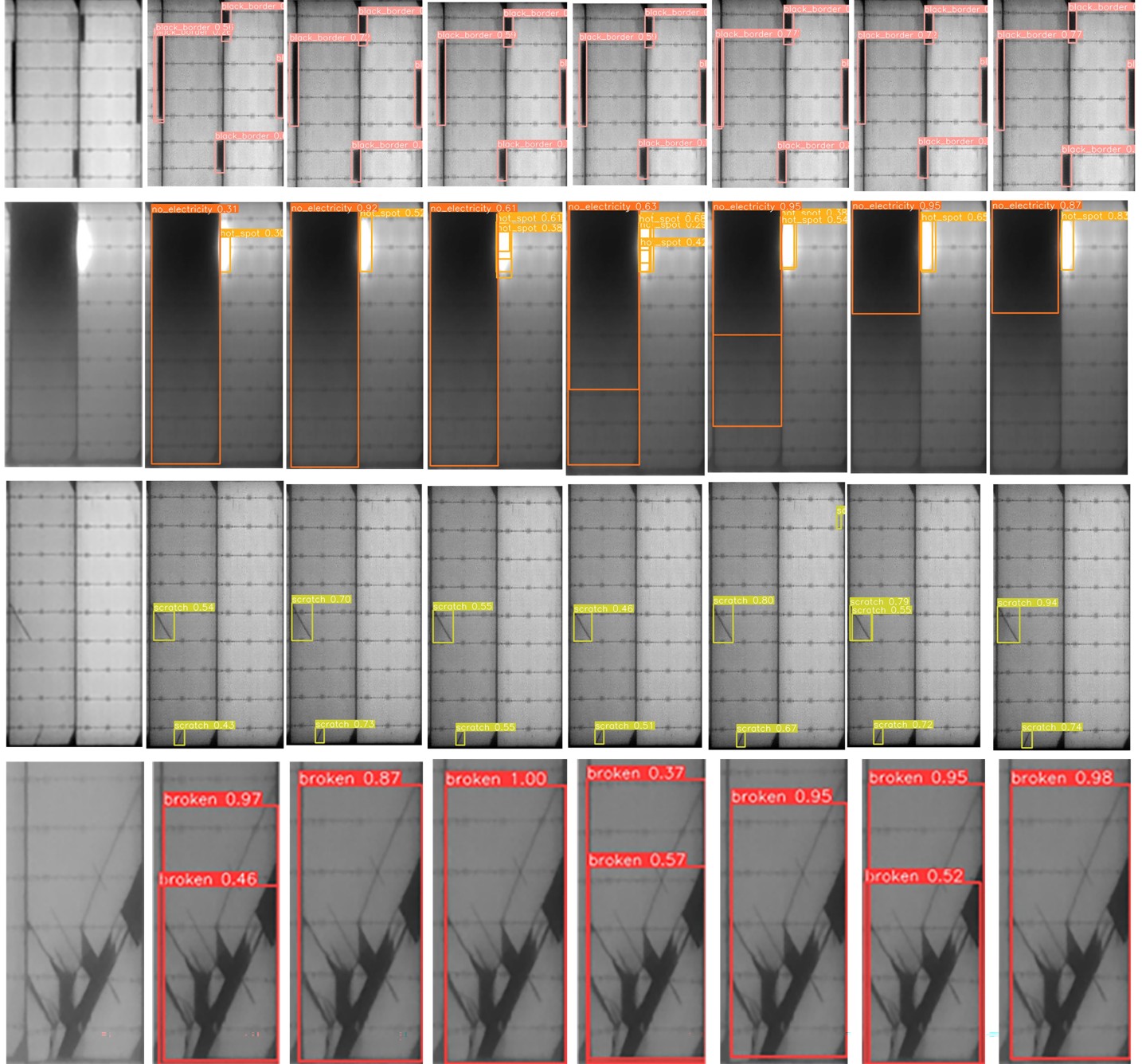

**Figure 8 Presentation of test results. From left to right: original image, YOLOv8n, MCSENet, GhostConv, BoTNet, Focaler-CIoU, data enhancement, our model.**

method was applied exclusively to the PVEL-AD dataset and the reverse-enhanced images, while the PVMD dataset was left unenhanced because its image quality is sufficiently high and does not require enhancement. I believe the primary focus of this article is to propose an improved YOLOv8-based defect detection model for photovoltaic panel

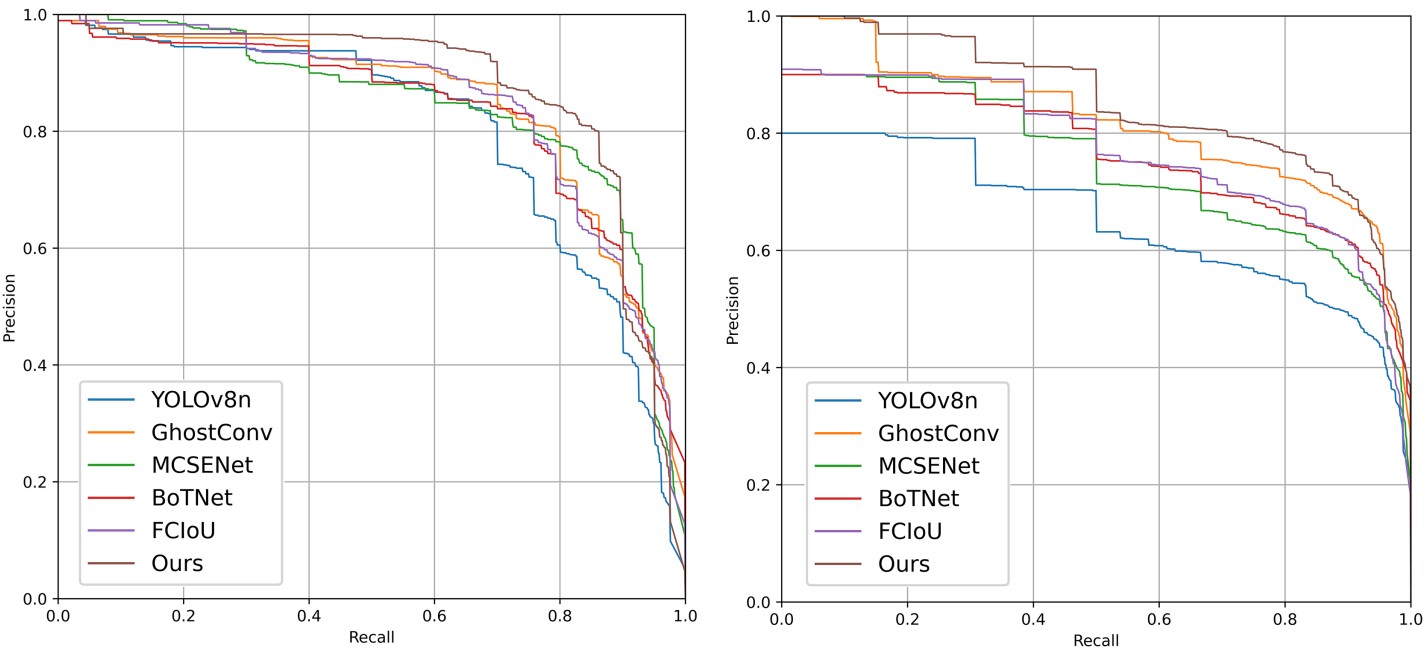

**Figure 9 Precision-recall (P-R) curves of each improved model on the PV-Multi-Defect-main (left) and PVEL-AD (right) datasets.**

electroluminescence defect images, rather than to introduce an image enhancement algorithm.

As illustrated in Fig. 10 and Table 7, the proposed model achieves significant improvements. These improvements arise because the baseline YOLOv8 model has limitations in feature extraction and bounding box localization, leading to horizontal and vertical misidentification of defects. Additionally, the defect bounding box annotations only cover part of the defect area, limiting effective detection. The introduction of GhostConv enhances feature extraction, allowing the model to more effectively capture subtle features at defect edges, thereby significantly improving detection accuracy. Figure 11 compares the performance of YOLOv8n and the proposed model in addressing horizontal and vertical misidentification of defect types. The precision-recall (P-R) curves for the improved models on the PVEL-AD dataset are shown in Fig. 9.

## Comparative experiment

In order to validate the advancements and innovations of the photovoltaic panel defect detection algorithm proposed in this article, we selected the PV-Multi-Defect-main dataset for testing under identical experimental conditions, without employing any dataset enhancement processing methods. The experimental environment and training parameters were consistent with those utilized in our proposed model, as detailed in Section "Environment and Training Parameters." The algorithms tested include RT-DETR (*Zhao et al., 2024*), versions v5–v8 of the YOLO series (*Linyi et al., 2023*), and SSD (*Liu*

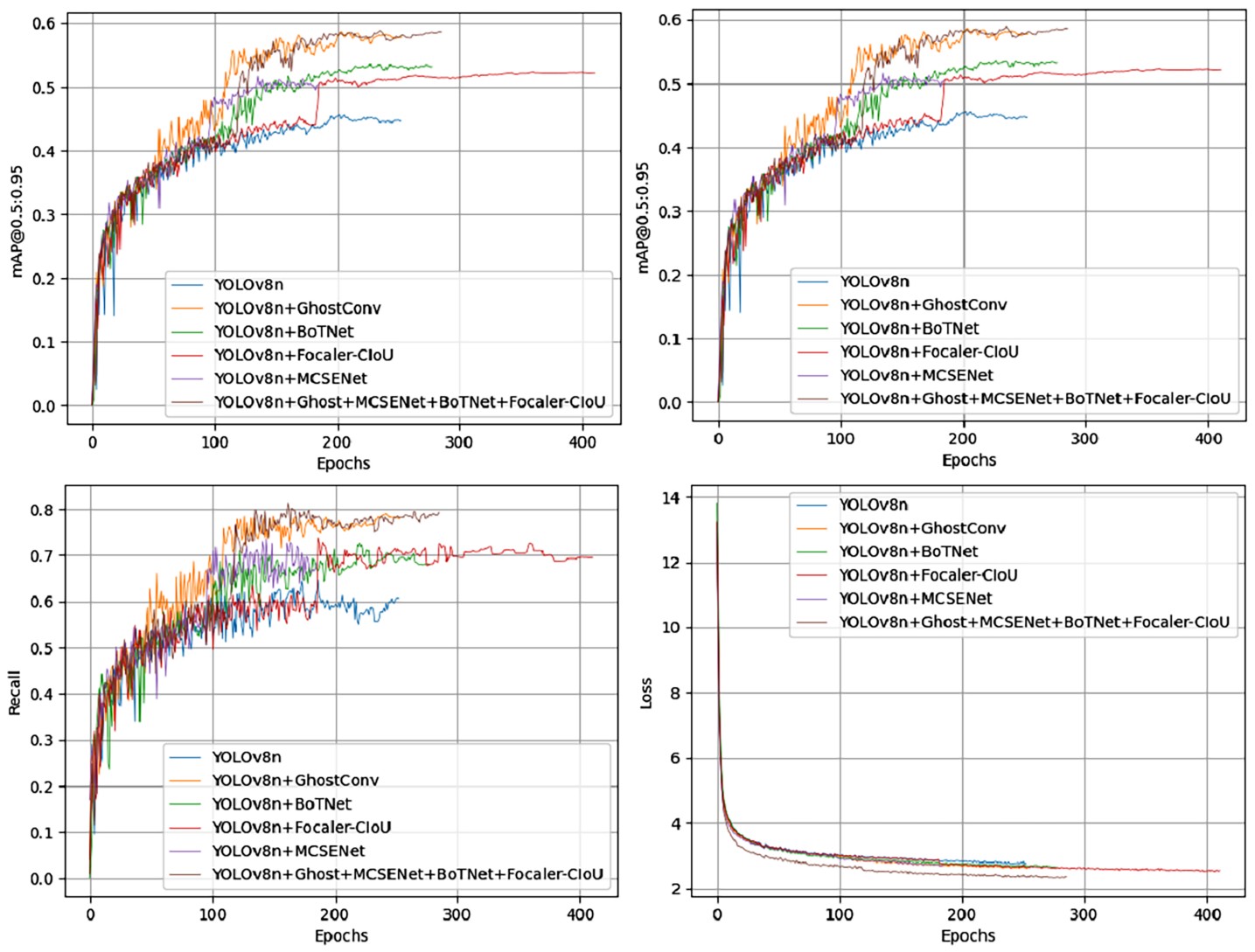

**Figure 10 mAP50, mAP50-95, recall, loss comparison of model test results on the PVEL-AD dataset.**

**Table 7 Comparison of the ablation results of PVEL-AD dataset.**

| Ghost Conv | BoT Net | MCSE Net | Focaler-CIoU | CycleGAN image enhance | mAP50 | mAP50-95 | Precision | Recall | Parameter (M) | GFLOPs (G) | FPS |
|---|---|---|---|---|---|---|---|---|---|---|---|
| | | | | | 0.645 | 0.457 | 0.797 | 0.588 | 6.4 | 8.2 | 214 |
| √ | | | | | 0.826 | 0.587 | 0.834 | 0.774 | 6.1 | 8.1 | 212 |
| | √ | | | | 0.760 | 0.536 | 0.723 | 0.708 | 6.8 | 8.4 | 194 |
| | | √ | | | 0.737 | 0.517 | 0.761 | 0.707 | 6.4 | 8.2 | 175 |
| | | | √ | | 0.762 | 0.524 | 0.747 | 0.726 | 6.4 | 8.2 | 218 |
| √ | √ | √ | √ | | 0.840 | 0.589 | 0.875 | 0.759 | 6.8 | 8.3 | 160 |
| √ | √ | √ | √ | √ | 0.935 | 0.680 | 0.883 | 0.899 | 6.8 | 8.2 | 184 |

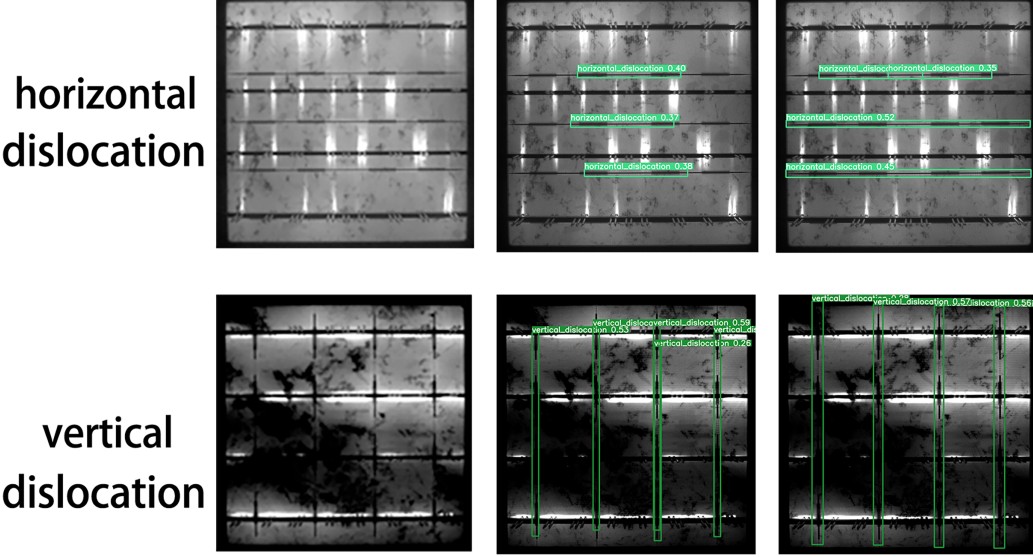

**Figure 11 Comparison of defect effects of horizontal and vertical error types, original picture (left), YOLOv8 (mid), improved model (right).**

**Table 8 Compare the experimental results with the different model.**

| Model | mAP50 | mAP50-95 | Precision | Recall | Parameter (M) | GFLOPs (G) | FPS |
|---|---|---|---|---|---|---|---|
| RT-DETR | 0.743 | 0.459 | 0.706 | 0.690 | 9,485,311 | 16.8 | 70 |
| SSD | 0.795 | 0.443 | 0.685 | 0.576 | 42,683,899 | – | – |
| YOLOv8n | 0.813 | 0.506 | 0.777 | 0.712 | 3,006,623 | 8.1 | 208 |
| YOLOv7n | 0.779 | 0.465 | 0.708 | 0.696 | 37,218,132 | 105.2 | 218 |
| YOLOv6n | 0.819 | 0.509 | 0.789 | 0.731 | 4,234,239 | 11.9 | 205 |
| YOLOv5n | 0.787 | 0.506 | 0.752 | 0.744 | 2,503,919 | 7.2 | 191 |
| Ours | 0.861 | 0.551 | 0.813 | 0.816 | 3,228,271 | 8.2 | 180 |

*et al., 2016*). The comparative results are presented in Table 8. From the table, it can be seen that the models YOLOv7n, RT-DETR, and SSD have a larger computational load and a bigger model size. The original models of YOLOv8n and YOLOv6n show better detection performance, achieving a balance between the number of parameters and detection accuracy. YOLOv5n has the smallest computational load, but its detection accuracy is relatively low. Compared to these object detection models, the improved detection model proposed in this article for photovoltaic panel defect electroluminescence images achieves the highest detection accuracy while maintaining a relatively low computational load. Additionally, Fig. 12 illustrates the P-R curve comparison of defect detection performance among various YOLO series target detection networks applied to the PV-Multi-Defect-main dataset.

From a comparison of these experimental results, it is evident that the photovoltaic panel defect detection model introduced in this article achieves commendable accuracy while maintaining a parameter count comparable to that of other models.

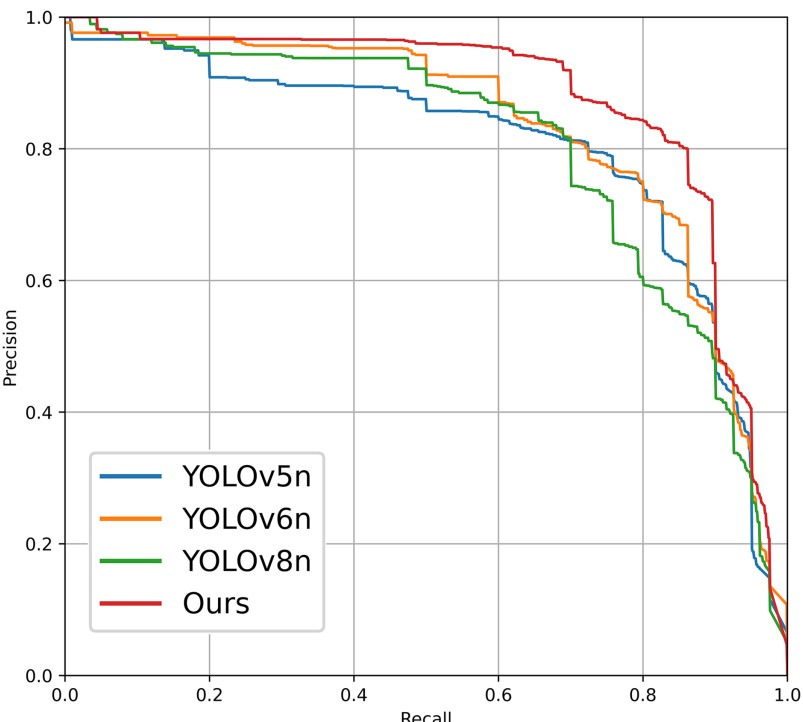

**Figure 12** P-R curves for different comparison schemes.

## CONCLUSIONS

In this article, a novel defect detection method for photovoltaic (PV) panels is proposed by improving the YOLOv8 baseline model. The research specifically addresses the challenges in accurately detecting defects in PV panels. A multi-channel squeeze-excitation network (MCSENet) is presented, which enhances channel representation through multi-channel squeeze-excitation operations, thereby improving the feature expression capability. The impact of GhostConv convolution on model accuracy and computational complexity is analyzed and tested, while the Bottleneck Transformer Network is introduced to enhance the feature extraction capabilities of the backbone network. Additionally, the Focaler-CIoU is incorporated into the loss function to replace CIoU, optimizing loss computation and improving detection accuracy. The proposed method also includes data augmentation techniques tailored to PV panel defect datasets.

The method is comprehensively evaluated on the PV-Multi-Defect-main and PVEL-AD datasets, and experimental results demonstrate that the proposed approach significantly enhances defect detection performance. Specifically, the method reduces false negatives and false positives, making it a robust and effective solution for PV panel defect detection. However, the findings also reveal certain limitations. For example, when MCSENet performs the squeeze-excitation operation, the extracted features may excessively dominate the original feature map during the fusion process, potentially diluting the original feature information. To mitigate this, the number of channels for SE

operations was empirically set to 3 in this study, which might have constrained the detection accuracy.

Therefore, future research should focus on improving the feature fusion process after each SE operation. By introducing a more balanced fusion mechanism—where the SE-extracted features and the original feature map are first merged and then scaled appropriately—it may be possible to reduce the overemphasis on extracted features, thereby further enhancing the feature extraction capability of MCSENet. Additionally, further exploration is needed to assess the practical deployment of this method in real-world PV panel production and maintenance scenarios, as well as its scalability to other defect detection tasks.

### Funding
This work was supported by the Research on Key Technologies of Regional Integrated Energy Multi-Energy Collaboration (20220203161SF), the design and verification of multi-value chain Collaborative data space management engine and management system architecture (2020YFB1707804). The funders had no role in study design, data collection and analysis, decision to publish, or preparation of the manuscript.

### Grant Disclosures
The following grant information was disclosed by the authors:
Research on Key Technologies of Regional Integrated Energy Multi-Energy Collaboration: 20220203161SF.
Collaborative Data Space Management Engine and Management System Architecture: 2020YFB1707804.

### Competing Interests
The authors declare that they have no competing interests.

### Author Contributions
- Jingdong Wang conceived and designed the experiments, performed the experiments, performed the computation work, prepared figures and/or tables, authored or reviewed drafts of the article, and approved the final draft.
- Zhu Cheng conceived and designed the experiments, performed the experiments, analyzed the data, performed the computation work, prepared figures and/or tables, and approved the final draft.

### Data Availability
The code is available in Zenodo: Cheng, Z. (2024). PV defect detection [Zenodo]. https://doi.org/10.5281/zenodo.13147569.
The PV-Multi-Defect dataset is available at GitHub and Zenodo:
- https://github.com/houhou34/PV-Multi-Defect.

- Cheng, Z. (2025). PV-Multi-Defect [Data set]. Zenodo. https://doi.org/10.5281/zenodo.15017563

The PVEL-AD dataset is available at: http://aihebut.com/col.jsp?id=118.

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
