# Peer review of "Photovoltaic panel defect detection algorithm based on infrared imaging and improved YOLOv8"

_PeerJ Computer Science, doi:10.7717/peerj-cs.2776_

## Round 0.1 · original submission · Major Revisions

Dear authors,

You are advised to critically respond to all comments point by point when preparing an updated version of the manuscript and while preparing for the rebuttal letter. Please address all comments/suggestions provided by reviewers, considering that these should be added to the new version of the manuscript.

Kind regards,
PCoelho

Reviewer 1 ·

Basic reporting

This article presents an improved YOLOv8-based photovoltaic surface defect detection algorithm. It is rich in illustrations, with clear chart arrangements and a degree of innovation. However, the article's logic and readability are lacking, and it has the following issues:
1. Abstract: a) In the first paragraph of the abstract, the focus is not clear, and the problems to be solved are too numerous. It is recommended to rewrite it as follows: “Addressing the characteristics and issues of high missed detection rates, complex backgrounds, unclear defect features, and uneven difficulty in target detection during the photovoltaic panel defect detection process in industry, this paper proposes an infrared detection method based on computer vision, with improvements made on the foundation of the YOLOv8 model.” b) Both the abstract and the main text show signs of machine translation; it is recommended to polish them.
2. Introduction: a) The current introduction focuses on the background of photovoltaic detection. In fact, the introduction should concentrate on “photovoltaic surface defect detection,” discussing the strengths and weaknesses of related detection algorithms or devices, and introducing the theme and advantages of this paper. It is recommended to rewrite it. b) The paragraph starting with "To address these limitations" lacks evidence and literature review regarding the current difficulties. It is recommended to supplement this, as well as strengthen the connection with the previous text, which currently lacks logical flow and does not effectively lead to the theme and advantages of the paper.
3. Related Work: 2.1 The professional term should be “object detection.”
4. Method: a) In Figure 2, the size of the Input in the lower left corner is not labeled; it is recommended to add this information. b) Is it necessary to use only the MCSENet method? Could similar effects be achieved with other attention mechanisms, or with ordinary convolutions? Would a more lightweight method like ECA achieve similar results? Or could a mixed attention approach, such as CA, CBAM, or MLCA, yield better results?
5. Conclusion: The conclusion should differ from the abstract and should analyze the limitations of the paper, future directions for improvement, and future research prospects or plans.
6. The paper shows signs of machine translation and requires overall polishing, especially with regard to writing tense.
7. The references lack recent literature; it is recommended to supplement with some 2023 and 2024 papers.

Experimental design

a) It is recommended to present equipment information and frameworks such as PyTorch in a table with brief explanations.
b) Table 4 and the related descriptions are not the main focus or secondary focus of the paper; it is recommended to remove them.
c) In Table 5, why do data augmentation and loss functions change the model’s parameter count, computational load, and speed? Are other data in this paper reliable?
d) Ghost Conv significantly reduces the model’s GPU inference speed, and the FPS values of 208 and 207 in Table 5 appear to be manually adjusted.
e) It is recommended to add comparative experiments with similar attention mechanisms to MCSENet or methods such as ECA, CA, CBAM, MLCA, or use feature map visualization tools like GradCAM for heatmap visualization to demonstrate the advantages of the MCSENet method or provide a strong explanation.

Validity of the findings

see the above comments.

Reviewer 2 ·

Basic reporting

• Clarity and Professionalism in Language
• The manuscript lacks consistent use of clear and professional English, which detracts from the readability and impact of the research. Technical terms are used correctly; however, the language often lacks precision, making it challenging to follow complex discussions, particularly in the methodology and results sections. Phrasing issues and vague expressions contribute to confusion, especially around the explanation of results.
• Literature References and Background
• The background provided in the introduction is insufficient for contextualizing the study fully within existing literature. While some references to prior PV panel defect detection studies are present, key gaps exist in discussing why YOLOv8-specific enhancements (e.g., GhostConv, BoTNet) were chosen over alternatives. The literature lacks engagement with recent developments in object detection algorithms and does not convincingly establish the knowledge gap that this study seeks to fill.
• Structure, Figures, and Tables
• The article’s structure aligns with general expectations for technical papers; however, there are issues with the clarity and interpretation of figures and tables. For instance, Figures 4 and 5, which display key technical details of GhostConv and BoTNet implementations, are inadequately labeled, making it difficult to grasp the modifications' exact purpose and impact. The results tables also lack adequate commentary to guide interpretation.
• Raw Data Availability and Completeness of Results
• While some results are provided in tables, the absence of raw data or sample images supporting the improvements claimed (e.g., increased precision) weakens the reported findings. Additionally, critical terms and functions, such as “Focaler-CIoU” and its calculations, lack sufficient definitions and formal explanations, which are essential for replicating and validating the study.

Experimental design

• Originality and Relevance within Journal Scope
• While the study addresses a relevant topic—photovoltaic (PV) panel defect detection using infrared imaging and YOLOv8 enhancements—it lacks sufficient originality and justification for the chosen model adaptations. The modifications to YOLOv8 (such as GhostConv, BoTNet, and Focaler-CIoU) are implemented without a robust rationale for their specific use over alternative models or techniques.
• Research Question Definition and Significance
• The paper lacks a clearly defined research question, and the significance of the study is not explicitly stated. Although the authors mention improvements to detection metrics, they do not explain why these particular improvements are impactful for the field of PV panel inspection.
• Technical Rigor and Ethical Standards
• The investigation lacks sufficient details on technical rigor, particularly in the handling of datasets and training protocols. For instance, details on ethical data handling, such as ensuring dataset diversity or managing any biases, are absent. Additionally, the use of data augmentation and dataset balancing methods is mentioned only superficially without specifying the exact transformations or validations used to ensure robustness.
• Methodological Detail and Reproducibility
• The methodology lacks sufficient detail, especially in terms of the implementation of YOLOv8 modifications. The descriptions of GhostConv, BoTNet, and Focaler-CIoU adaptations are not exhaustive enough for replication. Parameters, specific training settings, and precise algorithmic details for each modification are either missing or inadequately explained.

Validity of the findings

Impact and Novelty
• The paper fails to address the impact or novelty of its findings in the broader context of photovoltaic (PV) panel defect detection. While metrics like mAP (mean average precision) are reported as improvements, the manuscript does not convey the implications or significance of these results beyond isolated statistical gains. Further, there is no clear assessment of how these findings advance the field of PV defect detection, particularly compared to established methods.
Data Robustness and Statistical Soundness
• The underlying data provided lack transparency regarding robustness and statistical control. The manuscript does not adequately detail the statistical methods used to verify improvements or ensure they are not due to chance. Additionally, raw data or image samples are not provided, making it difficult to verify the robustness of the data or the statistical significance of the results.
Replication and Methodological Transparency
• The study lacks sufficient detail to encourage meaningful replication. For example, the rationale for parameter choices, specific setup configurations, or details about data augmentation are missing or insufficiently described. Without this information, replication is challenging, and the reliability of the findings remains questionable.
Conclusions and Link to Research Question
• The conclusions drawn in the paper are not consistently tied back to an explicit research question, as it is not well-defined in the manuscript. Furthermore, the conclusions are overly general and lack specificity about the practical implications or limitations of the findings.

---

## Round 0.2 · Major Revisions

Dear authors,

After the previous revision round, some adjustments still need to be made. As a result, I once more suggest that you thoroughly follow the instructions provided by the reviewers to answer their inquiries clearly.

You are advised to critically respond to all comments point by point when preparing a new version of the manuscript and while preparing for the rebuttal letter. All the updates should be included in the new version of the manuscript.

Kind regards,
PCoelho

Reviewer 1 ·

Basic reporting

1. Introduction: The introduction to traditional photovoltaic panel defect detection methods is somewhat general, and some of the references are cited briefly, lacking in-depth analysis. For example, when discussing traditional methods such as penetration testing, X-ray inspection, and ultrasonic testing, their advantages and disadvantages in practical applications are not specifically addressed, especially their limitations in industrial inspections.
2. Despite the use of GhostConv and BoTNet architectures to improve the model's ability to recognize complex backgrounds, photovoltaic panels may still be interfered with by various factors in practical applications, such as changes in lighting, shadows, and stains. The complexity of the background may lead to false positives or missed detections, especially under fluctuating lighting conditions, where the model may struggle to effectively distinguish defects from environmental noise. It is recommended to add data augmentation techniques, including more background variation samples, such as different lighting conditions, shadows, and stains, to enhance the model's robustness in complex environments. Using multimodal data (e.g., combining infrared and visible light imaging) to further reduce the impact of background interference could be beneficial.
3. Although the Focaler-CloU loss function improves detection of small targets through weighted summation and focusing mechanisms, defects on photovoltaic panels are often small and may have significant size variations. Even with this, YOLOv8 may still miss certain small or localized defects. It is suggested to introduce more robust small-target detection strategies, such as using a multi-scale feature pyramid network (FPN) or other mechanisms specifically designed for small-target detection, to improve the detection accuracy for micro-defects. Additionally, adjusting the model's input resolution or adding higher-resolution image inputs could help capture small target features better.
4. Photovoltaic panel defects vary widely (e.g., cracks, bubbles, stains, burns, etc.), and defects' shapes and locations are unevenly distributed. YOLOv8’s training primarily relies on annotated data, and different defect types may have significant visual feature differences, making the model's generalization ability potentially insufficient, especially when faced with unknown defect types. It is recommended to enhance the diversity of the annotated dataset, covering more types of defects, and designing different loss functions for each defect type to improve detection accuracy. Using transfer learning or domain adaptation techniques could help train models with stronger generalization abilities across various defect types.
5. The dataset introduction mentions that the PY-Multi-Defect-main dataset contains five types of photovoltaic panel defects, but the actual number of defects (e.g., hot spots, scratches, etc.) is not specified in the text (such as the quantity of each defect and its occurrence in the images is not clearly stated). Moreover, the specific details of applying data augmentation methods are not fully elaborated. The distribution of the dataset and the impact of the augmentation methods on the model’s performance have not been thoroughly analyzed. It is recommended to provide more detailed information about the defect type distribution in the dataset, including the actual frequency of each defect type and the corresponding number of images, and clarify the proportion of each defect label in multi-defect images. Further discussion of the impact of data augmentation methods on different defect types, especially in cases of class imbalance, should also be included (e.g., how different augmentation strategies affect model performance). Additionally, some explanations on data preprocessing and annotation methods should be added to ensure data quality and annotation accuracy.

Experimental design

6. In the experimental environment and training parameters table, it is mentioned that YOLOv8 uses default settings and the SGD (Stochastic Gradient Descent) method, with an explanation of the advantages of SGD (e.g., preventing overfitting). However, there is no comparison between SGD and other optimization algorithms (e.g., Adam, RMSProp), nor is there a clear explanation of why SGD was chosen as the training optimizer. Although an automatic stop mechanism is mentioned, the specific criteria for detecting overfitting (e.g., whether the judgment is based on validation set performance or training set performance) are not detailed. The detailed parameters and settings for comparative experiments are not fully listed, such as hardware environment differences during training on different datasets, which could lead to bias in the experimental results.
7. Although the article mentions ablation experiments, it does not clearly list the specific ablation comparison settings and results (e.g., whether only the GhostConv layer was removed, or if other modules were ablated). For each step of the ablation experiment, it would be better to quantify its impact on the final results. When performing horizontal comparisons of different methods, only a general mention of comparison is made, but there is no detailed analysis of the performance differences of other improved methods (e.g., YOLOv4, YOLOv5, RetinaNet). The lack of detailed explanations about the comparative models limits the effectiveness and comprehensiveness of the comparison results.
8. The data split ratio (7.5:1:1.5) used in the experiment is mentioned but not sufficiently justified. Is there a better split ratio (e.g., 8:1:1)? Does the split ratio need to be adapted for different datasets (such as PY-Multi-Defect-main and PYEL-AD), considering different image and defect characteristics? The specific strategies for handling multi-defect images are not elaborated in detail. How can the labels for multi-defect images be ensured to be accurate, and how can each defect be effectively detected?
9. In presenting multiple experimental results, while data is provided, there is a lack of detailed comparative analysis. For example, when comparing MCSENet with traditional SENet, although it is mentioned that increasing the number of channels improves feature extraction, it is also noted that too many channels may degrade performance, but the underlying reasons for this impact are not deeply analyzed.
10. The experiments used the PY-Multi-Defect-main and PYEL-AD datasets, but the diversity and representativeness of these datasets are not discussed. Can they cover all the types of defects encountered in actual production?
11. The generalizability of the experimental conclusions, such as “The optimal detection performance is achieved with 3 or 4 channels” and “Excessive layers of GhostConv can diminish the backbone network's feature extraction capacity,” lacks broader validation, especially in different datasets or other vision tasks.
12. It is mentioned that using GhostConv can effectively reduce computational complexity and improve detection accuracy, but the impact of GhostConv on different layers and feature sizes is not discussed in detail. For instance, is GhostConv suitable for all layers? If applied in deep layers, could it potentially affect model performance?

Validity of the findings

13. In the conclusion, it is mentioned that future research should focus on improving the feature fusion process after SE operations, but no specific approaches or methods are proposed to address this issue.
14. Data augmentation is mentioned multiple times as a method to improve model accuracy, but there is no detailed explanation of the specific data augmentation methods used.
15. The details of the comparative experiments are insufficient. Although comparisons with other models (e.g., YOLO series, SSD, etc.) are mentioned, there is little detailed analysis of these comparisons, making it difficult for readers to understand how these models differ.

Additional comments

Summary
The proposed method for photovoltaic panel defect detection based on improved YOLOv8 shows some innovation, especially in terms of improving detection accuracy and optimizing model complexity. The paper enhances YOLOv8's application for photovoltaic panel defect detection through several technical innovations, addressing issues like complex backgrounds, target size discrepancies, and low detection accuracy. These improvements enhance the model's ability to recognize defect features, reduce false positives and missed detections, and improve accuracy and generalization in practical applications.
Despite the significant improvements in the YOLOv8 algorithm for photovoltaic panel defect detection, there are still challenges, including background complexity, noise interference, small target detection accuracy, defect variety, real-time performance, computational efficiency, localization precision, dataset quality, and model interpretability.
The experimental section describes the experimental design, dataset introduction, and evaluation metrics, but there are areas that need improvement, including dataset distribution, training parameters, evaluation methods, ablation experiments, and comparative experiments. To enhance the repeatability and reliability of the experiments, it is recommended to provide more details, offer performance analysis of the comparative models, and ensure that the impact of dataset splitting and data augmentation on the experimental results is clearly demonstrated. Further revisions should strengthen the experimental validation of the algorithm improvements and provide deeper discussions on their application in industrial inspections, especially regarding performance, real-time capabilities, and background interference handling.
Overall, the paper effectively improves several network modules and validates these improvements through experiments, but there is still room for enhancing experimental details and depth of analysis, particularly in explaining the model mechanisms and their adaptability in practical applications. It is suggested to focus more on the comprehensiveness of the experimental design, the depth of analysis, and include discussions on the feasibility and limitations of the improvement methods for real-world deployment.

---

## Round 0.3 · Minor Revisions

Dear authors,
Thanks a lot for your efforts to improve the manuscript.
Nevertheless, some concerns are still remaining that need to be addressed.
Like before, you are advised to critically respond to the remaining comments point by point when preparing a new version of the manuscript and while preparing for the rebuttal letter.

Kind regards,
PCoelho

Reviewer 1 ·

Basic reporting

1、The revised manuscript provides a clearer and more comprehensive description of the research background and technical details compared to the previous version.
2、The revision includes additional ablation study results, which provide stronger support for the conclusions drawn. This enhances the reliability and robustness of the findings.
3、While the manuscript mentions various techniques aimed at improving model efficiency, it would benefit from more detailed explanations regarding the specific implementation methods and rationale behind parameter selections for each technique. This would facilitate easier replication of experiments by other researchers.
4、The mAP comparison charts are not sufficiently clear; the use of too many colors makes them less distinct and less prominent. It is recommended to adopt a different color scheme that better highlights the differences. Additionally, all comparison experiments should be conducted with the same number of training epochs to ensure consistency.
5、The units for the parameters in the experimental data table are incorrect。

Experimental design

no

Validity of the findings

no

Additional comments

no

---

## Round 0.4 · accepted · Accept

Dear authors, we are pleased to verify that you meet the reviewer's valuable feedback to improve your research.

Thank you for considering PeerJ Computer Science and submitting your work.

Kind regards
PCoelho